# Stabilization of Hydroxy-α-Sanshool by Antioxidants Present in the Genus *Zanthoxylum*

**DOI:** 10.3390/foods12183444

**Published:** 2023-09-15

**Authors:** Takahiko Mitani, Yasuko Yawata, Nami Yamamoto, Yoshiharu Okuno, Hidefumi Sakamoto, Mitsunori Nishide, Shin-ichi Kayano

**Affiliations:** 1Center of Regional Revitalization, Research Center for Food and Agriculture, Wakayama University, Wakayama 640-8510, Japan; 2Faculty of Education, Wakayama University, Wakayama 640-8510, Japan; namiyama@wakayama-u.ac.jp; 3Department of Material Science, Wakayama National College of Technology, Gobo 644-0023, Japan; okuno@wakayama-nct.ac.jp; 4Faculty of Systems Engineering, Wakayama University, Wakayama 640-8510, Japan; skmt@wakayama-u.ac.jp; 5Division of Food and Nutrition, Wakayama Shin-Ai Women’s Junior College, Wakayama 640-0341, Japan; nishide37m@gmail.com; 6Department of Nutrition, Faculty of Health Science, Kio University, Koryo-cho, Nara 635-0832, Japan; s.kayano@kio.ac.jp

**Keywords:** *Zanthoxylum piperitum*, hydroxy-α-sanshool, sanshools, α-tocopherol, stability, *N*-alkylamide, phenolic acid

## Abstract

Japanese pepper (sansho, *Zanthoxylum piperitum*) contains several types of sanshools belonging to *N*-alkylamides. Because of the long-chain unsaturated fatty acids present in their structure, sanshools are prone to oxidative deterioration, which poses problems in processing. In this paper, we evaluated the effects of antioxidants from the genus *Zanthoxylum* in preventing sanshool degradation using accelerated tests. An ethanolic extract of segment membranes of the sansho fruit pericarp was incubated at 70 °C for 7 days with different antioxidants to determine the residual amount of hydroxy-α-sanshool (HαS) in the extract. α-Tocopherol (α-Toc) showed excellent HαS-stabilizing activity at low concentrations. Among phenolic acids, we noted that the HαS-stabilizing activity increased with the number of hydroxy groups per molecule. For example, gallic acid and its derivatives exhibited excellent sanshool-stabilizing activity. Quercetin was found to be a superior HαS stabilizer compared with hesperetin and naringenin. However, the effective concentration was much higher for phenolic compounds than for α-Toc. These substances are believed to play a role in preventing the decomposition of sanshools in the pericarp of sansho. These sanshool stabilizers should be useful in the development of new beverages, foods, cosmetics, and pharmaceuticals that take advantage of the taste and flavor of sansho.

## 1. Introduction

The Japanese pepper (sansho, *Zanthoxylum piperitum*) has been used as a spice and a traditional herbal medicine (Kampo medicine) in Japan since ancient times. Wakayama in central Japan cultivates the variety Budo Sansho, which accounts for 60–70% of Japan’s sansho production [1]. The harvest of ripened sansho fruits is mainly conducted from the second half of July to the first half of August. After harvesting, the fruits are dried, and the seeds are removed. The resulting dried pericarps are preserved for later use. Half of the dried pericarps is utilized as spices, and the other half as Kampo medicine.

Sansho has a long history of being used as a spice, and in recent years, its pungent taste accompanied with numbness has led to a boom in its production. The pungent taste of Japanese peppers arises from sanshools. To date, 13 types of sanshools have been identified in the genus *Zanthoxylum* [2]. Among them, hydroxy-α-sanshool (HαS) is the main compound in Japanese and Chinese pepper (Huajiao, Sichuan peppercorn, *Zanthoxylum bungeanum*) [2,3,4,5,6]. Terpenes, such as citronellal, geraniol, and geranial, are the major fragrance components [7,8,9]. For using sansho as a spice or for preparing its extract, it is necessary to grind the pericarp. The pungency, aroma, and color tone of the dried pericarps of sansho are relatively stable at room temperature. However, when the pericarp is ground into a powder, the taste and aroma quickly disappear. Moreover, the intensity of the aroma and the tingling taste of Huajiao are not stable [9]. The tendency of sanshool to decompose when powdering sansho pericarp suggests that the pericarp contains substances or conditions that stabilize sansho. HαS has been reported to easily hydrolyze and oxidize under normal storage conditions [9,10]. Huajiao is often used in the form of chili oil, in which the pericarp is soaked in vegetable oil. The shelf life is expected to be relatively long because the sanshools gradually elute from the pericarp, and when they are placed in oil, they do not come in direct contact with air. However, further food processing is difficult in the presence of chili oil. If sanshools can be stabilized, sansho and huajiao could be used in a variety of processed and functional foods, and in pharmaceuticals.

Kampo medicines have been used traditionally for more than 1500 years in Japan. The sansho-containing Kampo medicines used in Japan are Daikenchuto (DKT), Chukenchuto, Tokito, and Shukubaito. Among these, DKT is one of the most frequently prescribed medicines for various digestive disorders [11,12]. It consists of Zanthoxylum Fructus (Japanese pepper), Zingiberis Siccatum Rhizoma (processed ginger), Ginseng Radix (Asian ginseng), and maltose powder [11]. In the other three Kampo medicines besides DKT, several herbs are blended, and apparently, the role of sansho is relatively small. The reasons for combining different herbal drugs are not fully understood and are being scientifically investigated. Unlike other herbal drugs, DKT has been studied to some extent. 6-Shogaol and 6-gingerol in ginger have a warming effect on the stomach and intestine, and ginseng compounds, such as ginsenoside, are believed to be effective in relieving fatigue, weakness, anorexia, and indigestion. Sansho has been shown to promote the secretion of gastrointestinal hormones, such as motilin, which may be linked to the promotion of gastrointestinal motility [13]. These findings suggest that sansho present in DKT plays a major role in gastrointestinal motility.

In addition to its gastrointestinal effects, strong salivating effects of HαS have been reported [14]. It is well known that HαS is involved in the activation of transient receptor potential cation channel subfamily V (TRPV), and the activation of somatosensory neurons through the inhibition of two-pore potassium channels (KCNK3, KCNK9, and KCNK18) [15,16,17]. Local anesthetic effects of HαS have also been studied [9]. HαS has also been reported to improve lipid metabolism in animal studies using rats and mice [18,19]. HαS can potentially be used as a therapeutic and health-promoting agent. Therefore, for further pharmacological research and development of its medicinal uses, sanshools need to be stabilized.

Sanshools are long-chain polyunsaturated amides. Therefore, it is conceivable that the instability of sanshools may be due to the oxidation of the polyunsaturated chain or amide hydrolysis under acidic conditions. We selected antioxidants reported to be present in the genus *Zanthoxylum*, and measured their stabilizing activity against sanshools using a screening system which we developed ourselves.

## 2. Materials and Methods

### 2.1. Sansho and Chemicals

*Zanthoxylum piperitum* fruits (variety name: Budo Sansho, product of Wakayama Prefecture, Japan) were harvested from Kainan, Wakayama, Japan. Ripened fruits were dried in a drier at 40 and 50 °C for 6 h each, and then at 60 °C for 2 h. Upon drying, the fruit split, and the pericarp was separated via sieving off the seeds. HαS and hydroxyl-β-sanshool (HβS) were obtained from MedChemExpress (Monmouth Junction, NJ, USA). Hydroxyl-ε-sanshool (HεS) was purchased from MuseChem (Fairfield, NJ, USA). Hydroxyl-γ-sanshool (HγS) was obtained from Chengdu Biopurify Phytochemicals Ltd. (Sichuan, China). High performance liquid chromatography (HPLC) grade acetonitrile, α-tocopherol (α-Toc), 2,2,5,7,8-pentamethyl-6-chromanol, phytol ((2E,7R,11R)-3,7,11,15-tetramethyl-2-hexadecen-1-ol), gallic acid monohydrate and ascorbic palmitate were purchased from Fujifilm-Wako Chemicals (Tokyo, Japan). 2-Hydroxybenzoic acid (salicylic acid), 4-hydroxy benzoic acid (PHBA), ethyl gallate, propyl gallate, 4-hydroxy-3-methoxybenzoic acid (vanillic acid), hydroxyquinone, and 3,4-dihydroxybenzoic acid (protocatechuic acid) were purchased from Tokyo Chemical Industry (Tokyo, Japan). Quercetin, naringenin, hesperetin, caffeic acid, *p*-coumaric acid, and ferulic acid were purchased from Sigma Aldrich (St. Louis, MO, USA). All other chemicals were of analytical grade.

### 2.2. Preparation of Segment Membrane

The pericarp of sansho mainly consists of flavedo and segment membranes (Figure 1). The segment membranes are located between the pericarp and seeds, and are attached to the inside of the pericarp. After separating the flavedo and segment membranes by placing the pericarp in a mortar and holding it down with a pestle, the membranes were peeled off from the pericarp using tweezers. The isolated segment membranes were ground in a laboratory mill (LM-PLUS, Osaka Chemical Co., Ltd., Osaka, Japan), and passed through a sieve with 250 μm pores to obtain the powdered segment membranes.

### 2.3. Accelerated Test of HαS-Stabilizing Activity

Three lots of sansho fruit were harvested from three sansho trees and dried separately. From the dried pericarps of the three lots (2 kg each), 200 g of pericarps were used for segmental membrane separation. Segment membranes (50 g each) were obtained. Segment membranes from each lot were powdered as described above. To perform the assay for one candidate compound in triplicate, segment membranes powder (0.4 g) was suspended in ethanol (8 mL) and mixed via vortexing. The suspension was centrifuged at 2000× *g* for 2 min, and the supernatant was collected. The amount of HαS in the alcohol extract of the segment membrane powder was determined using the HPLC method described below, and ethanol was added to bring the final HαS amount to 2.5 μmole/mL. Briefly, 2 mL of supernatant and 2 mL of the same volume of ethanol containing the candidate stabilizer were mixed in a 10 mL Spitz tube and incubated in an oil bath at 70 °C. The amount of HαS in the sample on day 7 of incubation was determined using HPLC. The ratio of the amount of HαS on day 7 of incubation to the amount on day 0 was calculated and expressed as survival rate (%). When ethanol without stabilizer was used, the amount of HαS remaining in the test solution was 61.35 ± 1.18% in the mean ± SD value. If it was above this value, it was judged to have a stabilizing effect, and if it was below, it was judged to have no stabilizing effect or to destabilize HαS. 

### 2.4. HPLC and LC-MS

HPLC was performed on a SHIMADZU LC-2010 instrument (SHIMADZU, Kyoto, Japan). All samples were injected into an InertSustainSwift C18 4.6 × 250 mm column (GL Sciences, Tokyo, Japan). The analytes were eluted with mixed solvent A (30% acetonitrile): B (80% acetonitrile) at a flow rate of 1.0 mL/min. The conditions were as follows: 0 min: 0% B; 35 min: 45% B; 50 min: 100% B; 51 min: 0% B; 55 min: 0% B. All analyses were performed at 40 °C and the absorbance of eluate was monitored at 270 nm. Liquid chromatography–mass spectrometry (LC-MS) was performed via electrospray ionization (ESI)-MS in a positive mode using a Bruker micrOTOF™ ESI-TOF mass spectrometer (Bruker Corporation, Billerica, MA, USA) interfaced with an Agilent 1200 HPLC system (Agilent Technologies, Inc., Santa Clara, CA, USA). The MS conditions are follows: capillary voltage, 4500 V; drying gas temperature, 200 °C; drying gas flow, 8.0 L/min; nebulizer pressure, 1.6 Bar. A portion of the sample was loaded onto an Inertsil ODS-3 (2.1 × 150 mm) column. Separation was achieved via elution with a mixture of solvents A (30% acetonitrile) and B (80% acetonitrile) at a flow rate of 1.0 mL/min. The conditions were as follows: 0 min: 0% B; 35 min: 45% B; 50 min: 100% B; 51 min: 0% B; 55 min: 0% B. All analyses were performed at 40 °C, and the absorbance of eluate was monitored at 270 nm.

### 2.5. Quantitation of HαS

HαS of the samples listed in Materials and Methods were diluted to 0.38, 0.76, 1.90, and 3.8 mmole/mL, respectively, using an ethanol solution containing 0.1% α-Toc, and analyzed using HPLC (*n* = 3) under the conditions described above. In this concentration range, a linear relationship was obtained between the HαS concentration and the HPLC peak areas. This calibration curve was used to determine the HαS content in the sample. The experiment was also performed three times on different days, but no daily difference was found. The limit of detection in this method was 0.00163 µmole/mL.

### 2.6. Quantitation of Tocopherol

Analysis of α-, β-, γ-, and δ-Tocs was performed using HPLC by Japan Food Research Laboratories (Suita, Osaka, Japan). The method is shown below. NaCl (1%, 1 mL), 6% pyrogallol in ethanol (10 mL), ethanol (1 mL) incorporating 2,2,5,7,8-pentamethyl-6-hydroxychroman (PMC; 10 mg), and 60% KOH (1.5 mL) were added to the crushed pericarp (0.5 g) or seed (1 g). The mixture was kept at 70 °C for 30 min. After cooling in an ice–water mixture, the unsaponified compound was extracted three times with 1% NaCl (15 mL) and 1:9 ethyl acetate/hexane (15 mL). The organic solvent layer was combined and evaporated to dryness under vacuum [20]. The residue was dissolved in hexane (1 mL) and analyzed via HPLC. HPLC was performed on a SHIMADZU LC-20AT instrument with fluorescence detector RF-10AXL. All samples were injected into an YMC-Pack SIL-06 S-5 μm, 4.6 × 150 mm column (YMC Co., Ltd., Kyoto, Japan) for α and δ-tocopherol, and YMC-Pack SIL-06 S-5 μm, 4.6 × 250 mm for β and γ-tocopherol. The analytes were eluted with acetic acid/n-hexane/2-propanol (5:100:2 *v*/*v*/*v*) at a flow rate of 1.5 mL/min at 40 °C. The excitation and emission wavelengths were 298 and 325 nm, respectively.

### 2.7. Statistical Analysis

All data are expressed as the mean ± standard deviation (SD) values for each group. Statistical analysis was performed using one-way analysis of variance with Student’s *t*-test. *p* < 0.05 and are indicated by a single asterisk (*) or different letters.

## 3. Results

### 3.1. Sansho Harvest Season

Sansho fruits are harvested in three seasons in the central prefecture of Wakayama, Japan (Table 1). The unripe fruits are harvested in May. The seeds of these fruits have not yet hardened, and the whole fruit is used for Tsukudani (a preservable food, boiled in soy sauce). Ripe fruits are harvested from the second half of July to the first half of August. After drying, they are used as raw materials for herbal medicines and spices. In September, the sansho fruit turns red and is harvested as fully ripened sansho pepper, which is also used as a spice. However, in Wakayama Prefecture, the amount of fully ripened fruit harvested is extremely small. Therefore, ripened fruits were used in this study.

### 3.2. HPLC and LC/MS of Ethanol Extracts of Pericarps

A chromatogram of the ethanolic extract of pericarp is shown in Figure 2a. Several peaks were observed in the chromatogram. Of these, peaks A, B, C, and D were found to be of HεS, HαS, HβS, and HγS, respectively, after comparison with the standard sanshools. In addition, MS analysis was performed for peaks B, E, and F. Peak B compound had the elemental composition of C16H25NO2 due to the signals of 264.1958 (M + H^+^) and 286.1778 (M + Na^+^) based on high-resolution LC-MS (ESI, positive mode) data. Similarly, we found that peak E compound had the elemental composition of C16H25NO due to the signals of 248.2009 (M + H^+^) and 270.1828 (M + Na^+^), and peak F compound had the elemental composition of C18H27NO due to the signals of 274.2165 (M + H^+^) and 296.1985 (M + Na^+^). Therefore, peaks B, E, and F were believed to be for HαS, α-sanshool (αS), and γ-sanshool (γS), respectively. From HPLC of references performed under the similar conditions [14,21,22], we assumed that peaks E and F might be for αS and γS, respectively.

### 3.3. Effect of Alcohol Concentration on Extraction Amount of HαS and Its Stability

The pericarp powder (0.4 g) was suspended in aqueous ethanol solutions of various concentrations (20 mL). After vigorous mixing for 5 min at about 20 °C, the mixture was centrifuged at 2000× *g* for 2 min, and the supernatant was recovered. The amount of HαS in each supernatant was measured using HPLC. As shown in Figure 3, the amount of HαS extracted was constant when the ethanol concentration was up to 50%, but decreased with a decrease in ethanol concentration to 50% or less. Furthermore, when these extracts were incubated at 70 °C, the instability of HαS increased with the increasing proportion of water in the eluate (Figure 4). These results indicated that the HαS stabilizer, thought to be present in the pericarp, is soluble in ethanol.

### 3.4. Comparison of the Stability of HαS in Ethanolic Extracts of Pericarp and Segment Membranes

HαS stabilizer, found to be present in the sansho pericarp, was apparently extracted with ethanol. Therefore, to search for stabilizing substances, we considered it necessary to build a system that minimized contamination of stabilizers derived from sansho. We collected segment membranes attached to the inside of the pericarps and prepared their ethanolic extract. The amount of HαS in the segment membrane was approximately 5 mg/g dry weight, which was equivalent to approximately 1/20 of that in the pericarp. As shown in Figure 2b, the HPLC chromatogram of the ethanolic extracts of segment membranes was similar to that of the ethanol extract of the pericarp. 

The two ethanolic extracts were incubated at 60, 70, and 80 °C for 1 week and the residual amounts of HαS in them was compared (Figure 5). The amount of HαS in 95% ethanolic extract of the segmented membranes was reduced to 61.5 ± 4.65% in 7 days at 70 °C. On the contrary, the amount of HαS in the ethanolic extract of pericarps was maintained above 90% at 70 °C. Furthermore, because the segment membranes contain less chlorophyll and polyphenols, the influence of these substances on the assay system is believed to be weakened. Therefore, by adding various candidate antioxidants to the ethanolic extract of the segment membranes, we investigated their HαS-stabilizing activity.

### 3.5. HαS-Stabilizing Activity of Antioxidant Vitamin

Tocopherol is a fat-soluble antioxidant vitamin. Hisatomi et al. reported that the α-Toc content in the pericarp of sansho is 35 mg/100 g, which is approximately 11.7 times that in the seeds [23]. This vitamin E content belongs to a food ranked very high in the Standard Tables of Food Composition in Japan. In the pericarp of Budo sansho, we found that α-Toc content was 10.7 mg/g of dry weight and that β-, γ-, and δ-tocopherol were not present. Using a screening system, we attempted to examine the HαS-stabilizing activity of α-Toc. As shown in Figure 6, α-Toc exhibited a superior HαS-stabilizing activity even at a low concentration. As α-Toc contains a phenolic-chromanol ring linked to a saturated isoprenoid side chain (phytol), we also examined the HαS-stabilizing activity of 2,2,5,7,8-pentamethyl-6-chromanol and phytol. The chemical structure of each compound is shown in Figure 7. 2,2,5,7,8-Pentamethyl-6-chromanol showed excellent HαS-stabilizing activity, but phytol did not show any such activity (Figure 6). It is suggested that the antioxidant activity of 2,2,5,7,8-Pentamethyl-6-chromanol in α-Toc is related to the HαS-stabilizing activity. β-carotene is a red-orange pigment that is abundant in plants and has antioxidant properties. Carotenoids are expected to be present in the pericarps of sansho and may contribute to the stabilization of sanshools. However, it is difficult to dissolve in ethanol, and even if it did dissolve, it is unstable by itself, making its application in food products difficult. This is an issue for future studies.

Ascorbic acid is a well-known water-soluble antioxidant vitamin normally present in plants. However, because it is insoluble in ethanol, the HαS-stabilizing activity of ascorbic acid could not be investigated using our assay system, and we believe that it is not a stabilizer of HαS. Ascorbic palmitate is an ester formed from ascorbic acid and palmitic acid, which is hydrolyzed into fat-soluble ascorbic acid and is used as a food additive. We investigated the HαS-stabilizing activity of ascorbic acid using ascorbic palmitate. It showed the highest HαS-stabilizing activity at approximately 2 mM, and the activity decreased at higher concentrations (Figure 6). 

### 3.6. HαS-Stabilizing Activity of Phenolics

Many phenolic acids, including hydroxycinnamic acid, flavonoids, lignans, and coumarins, have been reported to be present in the genus *Zanthoxylum* [23,24,25,26,27,28]. We investigated the HαS-stabilizing activity of several of these compounds using our screening system. The HαS-stabilizing activities of mono-hydroxybenzoic acids (salicylic acid, PHBA, and vanillic acid), di-hydroxybenzoic acid (protocatechuic acid), and tri-hoxybenzoic acid (gallic acid, ethyl gallate, and sodium gallate) are shown in Figure 8. The chemical structures of hydroxybenzoic acids and their derivatives are shown in Figure 9.

Gallic acid and its related compounds showed high HαS-stabilizing activity at low concentrations (Figure 8). This was followed by protocatechuic acid, PHBA, and vanillic acid. As the number of phenolic hydroxy groups per molecule increased, the HαS-stabilizing activity increased. Among the mono-hydroxybenzoic acids, there was a difference in the HαS-stabilizing activity. The acidity of each of the three selected compounds differed; the pKa values of salicylic acid, PHBA, and vanillic acid are 2.97, 4.54, and 4.51, respectively. The acidity of mono-hydroxybenzoic acids appeared to affect the stability of HαS. Butylated hydroxyanisole (BHA) and butylated hydroxytoluene (BHT) are synthetic lipophilic compounds, chemically derived from phenol. Owing to their antioxidant properties, they have been widely used as preservatives in edible oils. Because there is only one phenolic hydroxyl group in these two compounds, the sanshool-stabilizing effect is weak.

Hydroxycinnamic acids are widely distributed in plants. The HαS-stabilizing activity of caffeic acid was found to be higher than that of *p*-coumaric and ferulic acids (Figure 8). Caffeic acid has two phenolic hydroxyl groups, whereas *p*-coumaric acid and ferulic acid only have one each (Figure 10). Flavonoids are also widely distributed in plants and have been reported from the genus *Zanthoxylum* [23,24,25,26,27,28]. We examined the HαS-stabilizing activities of quercetin, naringenin, and hesperetin (Figure 10). Quercetin showed a high HαS-stabilizing activity at a concentration of ≥0.1 mM (Figure 8). The HαS-stabilizing activity of quercetin, which has five hydroxyl groups in its structure, exceeded those of naringenin and hesperetin with two hydroxyl groups each. Arbutin, a sugar-containing hydroquinone, has been reported to be present in *Zanthoxylum piperitum* [23]. Considering the insolubility of arbutin in ethanol, we did not attempt to investigate its HαS-stabilizing activity. Hydroquinone, the aglycone portion of arbutin, has two hydroxyl groups attached to the benzene ring at the para-position and may act as a reducing agent. In preliminary tests, hydroquinone showed weak HαS-stabilizing activity.

### 3.7. Effect of pH on the Stability of HαS

Ethanolic extracts of segmented membranes were mixed with the same volume of Britton–Robinson buffer (pH 2.2–12) [29], and the mixtures were incubated at 70 °C. On days 3 and 9, the HαS content in each mixture was assayed. A significant decrease in the HαS content was observed with sanshools degrading under acidic conditions (Figure 11).

## 4. Discussion

In this study, several compounds with stabilizing activity for HαS, which is the main pungent substance sansho, were found. Among these, α-Toc was found to substantially stabilize HαS in sansho pericarp. This appears to be one of the reasons for the abundance of α-Toc in sansho pericarp, which is an amphiphilic compound present in cell membranes. α-Toc, which is synthesized in chloroplasts and is present in cell membranes, plays an important role in preventing lipid oxidation [30,31]. The three-dimensional arrangement of α-Toc in lipid bilayer membranes in cell membranes has been studied [32,33]. Sanshools are amphiphilic compounds with fat-soluble fatty acid side chains and a hydrophilic amide moiety. The two compounds share common structural features. We speculate that sanshools may be present in the cell membrane or chloroplast lamellae of pericarp cells in close proximity to α-Toc. Further research on the interaction between α-Toc and sanshools and their locations in the cells is required. Since α-tocopherol is widely used as a safe antioxidant food additive in many countries, it may be a useful substance for stabilizing sanshools.

Some phenolic compounds reported in *Zanthoxylum* exhibited high HαS-stabilizing activity. However, because these compounds are believed to be bound to sugars and organic acids, which makes them hydrophilic, they are difficult to access HαS. Therefore, we believe that they are not involved in the stabilization of HαS in the pericarp. Furthermore, the concentration of phenolic compounds, which are effective in stabilizing HαS, was much higher than that of α-Toc; it may, therefore, be difficult to practically use these phenolic compounds. Furthermore, as shown in Figure 8, high concentrations of gallic acid derivatives are required for the stabilization of HαS. Quercetin is effective even at low concentrations; however, because it is not on the list of food additives as an antioxidant, we did not conduct further studies on the mechanism of action of gallic acid derivatives and quercetin. We intend to investigate these mechanisms in a future study.

We believe that the mechanism by which antioxidants stabilize HαS is the same as that involved in the prevention of oxidation of vegetable oils rich in unsaturated fatty acids. In this case, the antioxidants do not bind to the unsaturated fatty acids at a specific site through a special binding, such as that seen for enzyme–substrate binding. The antioxidants are instead believed to react with oxygen radicals present in the environment surrounding the unsaturated fatty acid, reducing the frequency with which oxygen radicals approach the unsaturated fatty acid. 

HαS has also potential use for the treatment of diseases and in healthcare. Therefore, for further pharmacological research or development of the medicinal uses of sanshools, it is important to identify the conditions or substances that stabilize sanshools. Methods for the synthesis of sanshools have also been reported [34,35,36], and the pure products, thus obtained, can be used; however, their stability is likely to be even more important. 

Echinacea is a popular herbal medicine in Western countries that has been used to treat colds and influenza [37,38]. However, its usefulness in clinical trials has been questioned [38]. There are various reasons for this. Liu and Murphy suggested that the stability of *N*-alkylamide, which is one of the active ingredients of Echinacea, poses a problem, and its content is not guaranteed for each preparation [39]. *Echinacea purpurea* contains eight isobutylamides, all of which are susceptible to oxidative degradation and are very similar to sanshools. They found that phenolic acids in Echinacea prevented the degradation of these isobutylamides depending on the experimental conditions [39]. It is believed that the degradation of *N*-alkylamides is prevented to some extent by the antioxidant action of these phenolic acids. However, because *N*-alkylamides are vulnerable to acids, further stability can be expected using α-Toc.

## 5. Conclusions

We constructed a system for stabilizing substances in sanshools and found that α-Toc exhibits excellent activity. Stabilization of sanshools can be employed for developing new sansho products that exploit their excellent functionality. However, since sanshools and α-Toc are not strongly bonded, they are likely to separate easily when turned into liquids or powders. In the future, it will be necessary to develop a technology to make sanshools and α-Toc always coexist in close proximity.

## 6. Patents

TM is named an inventor on a patent (Takahiko Mitani and Takashi Tsuchida, Patent JP, 6630880, B).

## Figures and Tables

**Figure 1 foods-12-03444-f001:**
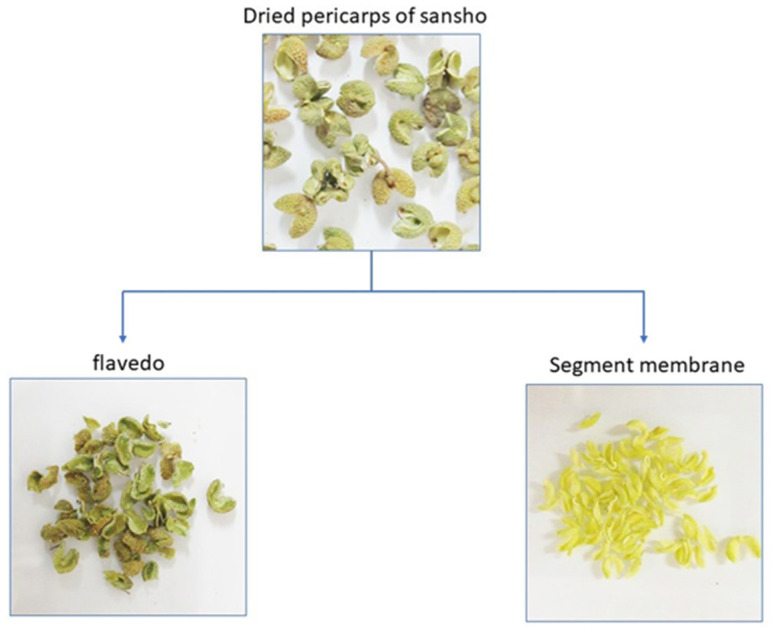
Preparation of flavedos and segment membranes from dried pericarps. The flavedo and segment membranes were separated by holding the dried sansho pericarp in a mortar with a pestle.

**Figure 2 foods-12-03444-f002:**
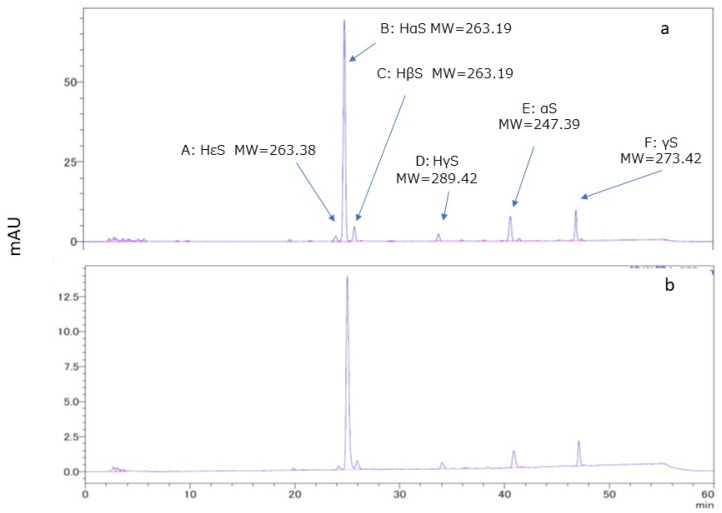
HPLC and LC-MS of ethanolic extracts of pericarp and segment membranes. The retention times (RT) for peaks A–D and the sanshool standards were compared, and the respective compounds were identified. The molecular weights of the compounds represented by peaks B, E, and F were determined using LC-MS analysis. (**a**), ethanolic extracts of pericarps; (**b**), ethanolic extracts of segment membranes.

**Figure 3 foods-12-03444-f003:**
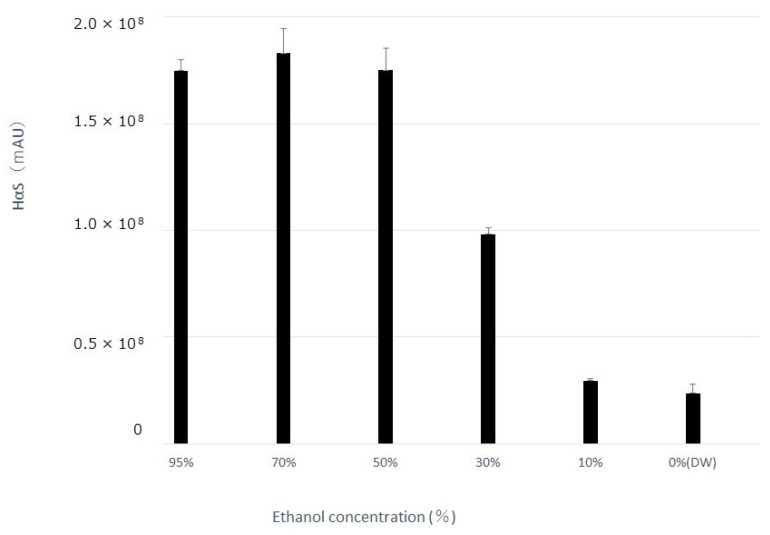
Effects of differences in ethanol concentration on the extraction of HαS. Sansho pericarp powder (0.4 g) was suspended in solvent (20 mL) having different ethanol concentrations, mixed vigorously, and centrifuged. The content of HαS in the supernatant was analyzed using HPLC.

**Figure 4 foods-12-03444-f004:**
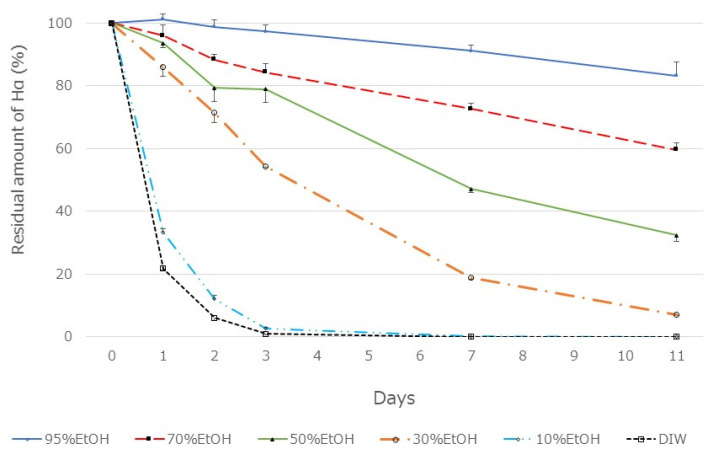
Thermal stability of HαS in sansho pericarp extract obtained using a solvent with different ethanol concentrations. Sansho pericarp powder (0.4 g) was mixed in solvent (20 mL) having different ethanol concentrations, mixed vigorously, and centrifuged. The supernatant was incubated at 70 °C for 11 days. On days 1, 2, 3, 7, and 11, the amount of HαS in the samples was analyzed using HPLC.

**Figure 5 foods-12-03444-f005:**
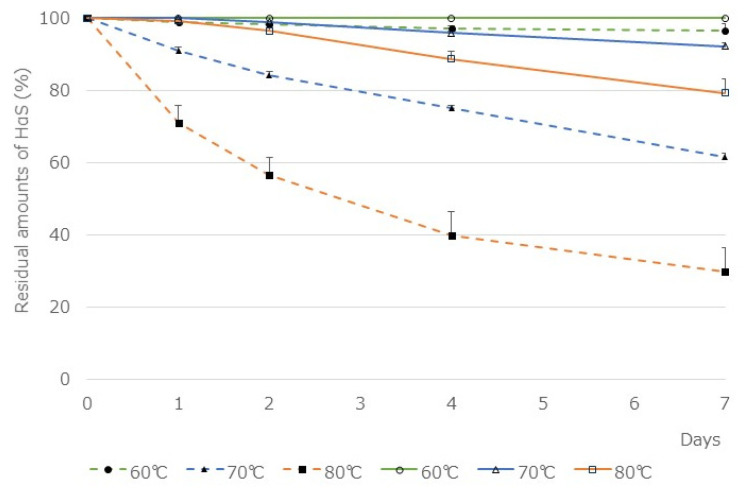
Thermal stability of HαS in ethanolic extracts of pericarp and segment membranes. Ethanolic extracts of sansho pericarp and segment membranes were prepared as described in the Section 3.3. Each extract was incubated for 1 week at 60, 70 or 80 °C, and the amount of HαS in the samples was analyzed using HPLC. The solid line shows the ethanolic extract of the pericarp and the dashed line shows that of the segment membranes.

**Figure 6 foods-12-03444-f006:**
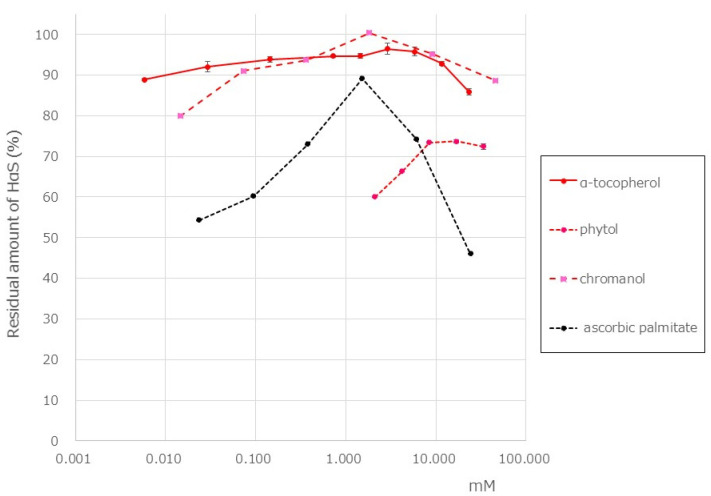
Effect of antioxidant vitamins on the stability of HαS at 70 °C. The ethanolic extract of sansho segment membranes was prepared as described in Materials and Methods. The candidate antioxidant compounds were dissolved in ethanol at various concentrations. Both the ethanolic solutions were mixed and incubated for 1 week at 70 °C. The amount of HαS in the samples was analyzed using HPLC.

**Figure 7 foods-12-03444-f007:**
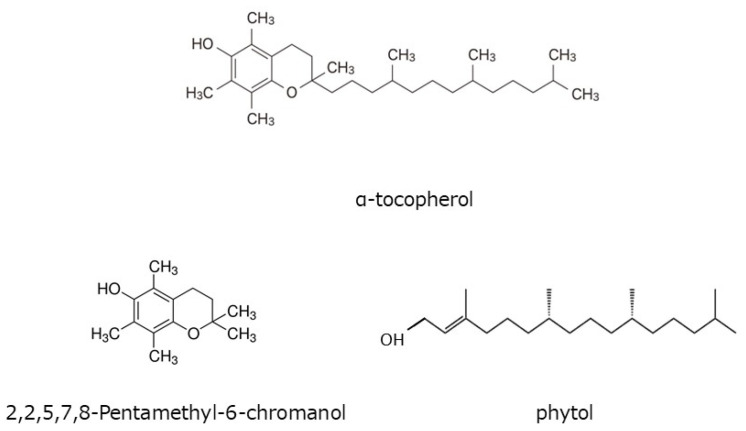
Chemical structure of α-tocopherol, 2,2,5,7,8-pentamethyl-6-chromanol, and phytol.

**Figure 8 foods-12-03444-f008:**
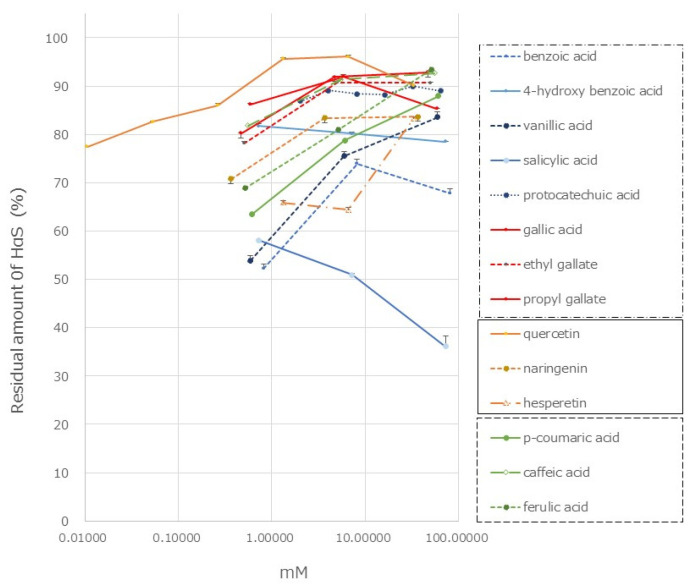
Effect of phenolics on the stability of HαS at 70 °C. The ethanolic extract of sansho segment membranes was prepared as described in Materials and Methods. The candidate compounds were dissolved in ethanol at various concentrations. Both the ethanolic solutions were mixed and incubated for 1 week at 70 °C. The amount of HαS in the samples was analyzed using HPLC.

**Figure 9 foods-12-03444-f009:**
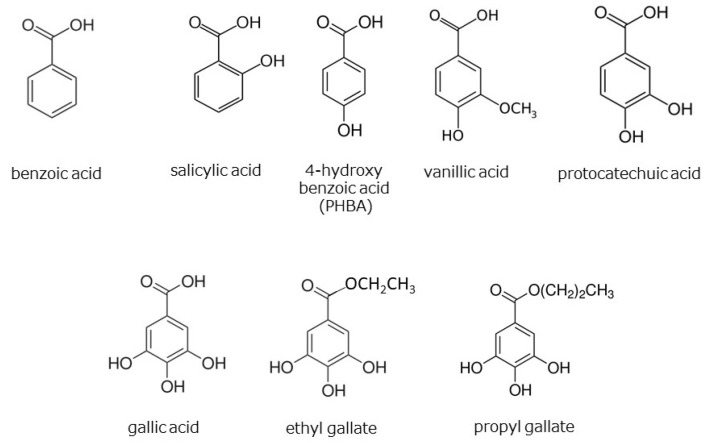
Chemical structures of hydroxybenzoic acids and their derivatives.

**Figure 10 foods-12-03444-f010:**
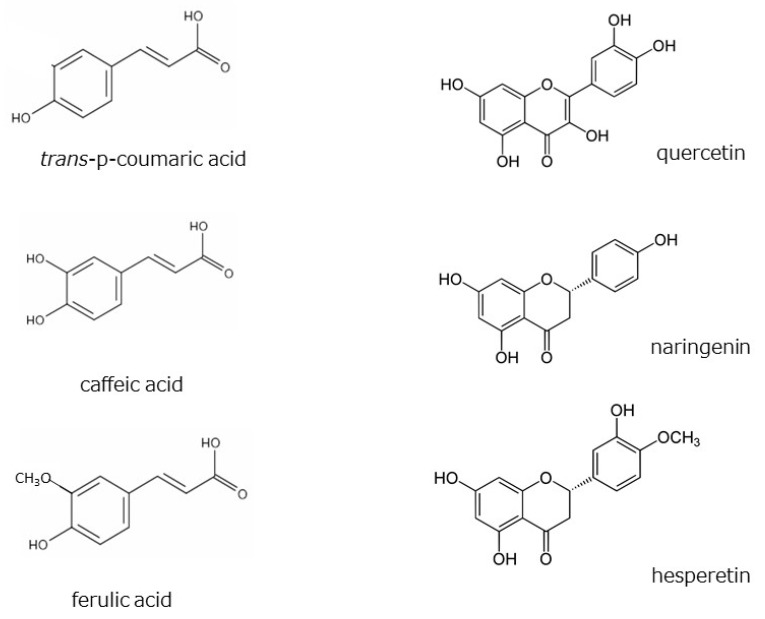
Chemical structures of hydroxycinnamic acids and flavonoids.

**Figure 11 foods-12-03444-f011:**
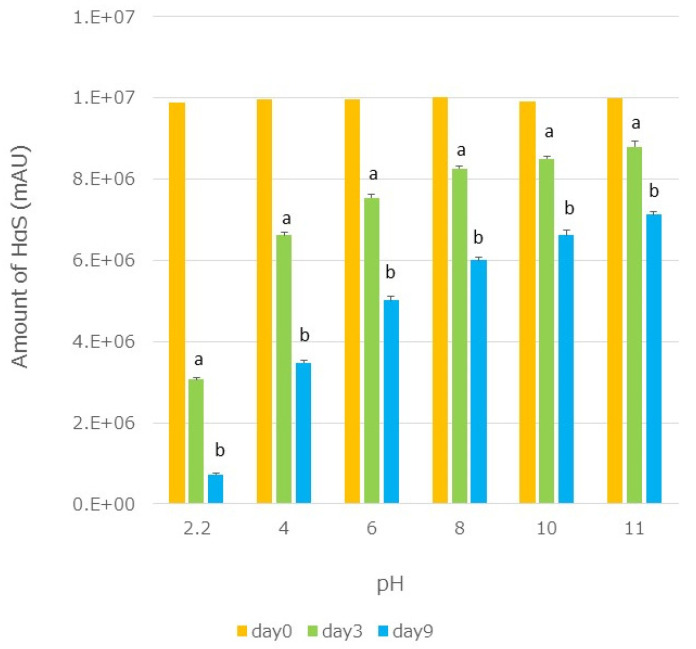
Effect of pH on the stability of HαS. Ethanolic extracts of segmented membranes were mixed with the same volume of Britton–Robinson buffer (pH 2.2–12), and the mixtures were incubated at 70 °C. On days 3 and 9, the HαS content of each mixture was assayed. *p*-values < 0.05 were considered statistically significant and are indicated by different letters.

**Table 1 foods-12-03444-t001:** Features of sansho fruits at different maturation stages.

Stage	Harvest Time	Color of the Pericarp	Seed
Unripe fruits	May	Green	Unmatured, soft, pale green
Ripe fruits	Late July to Early August	Pale green	Mature, black
Fully ripened fruits	September onward	Pale red to red	Mature, black

## Data Availability

The datasets generated for this study are available on request to the corresponding author.

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
