# Peer review of "Stabilization of Hydroxy-α-Sanshool by Antioxidants Present in the Genus Zanthoxylum"

_foods, 2023, doi:10.3390/foods12183444_

Round 1

Reviewer 1 Report

The work is well designed and well written. However, the following suggestion may be considered:

1. In introduction provide a description on the herbal medicines(a list is acceptable) in which the pericarp is used and the reason for its incorporation (property)

2. The statistical tools used in the analysis of the results is not mentioned in the materials and methodology section. 

3. Is there any possibility to elucidate the underline mechanism by anti-oxidants like gallic acid and quercetin to stabilize HαS? If yes, why it is not included?

4. At what quantity the antioxidants were mixed with HαS?

5. The possibility to have an alteration in the biological activities of HαS cannot be ruled out. In such case why the biological activities not estimated after stabilizing HαS with anti-oxidants?

6. the introduction should also include  small section on biological activities of hydroxy-α-sanshool.

Manuscript is well written and some minor grammar check is required.

Author Response

Responses to the reviewer’s comments

  1. In introduction provide a description on the herbal medicines (a list is acceptable) in which the pericarp is used and the reason for its incorporation (property)

As suggested by the reviewer, we have added the following text on herbal medicines in which the pericarp is used and described their properties:

Kampo medicines have been used traditionally for more than 1,500 years in Japan. The sansho-containing Kampo medicines used in Japan are Daikenchuto (DKT), Chukenchuto, Tokito, and Shukubaito. Among these, DKT is one of the most frequently prescribed medicines for various digestive disorders [10,11].  It consists of Zanthoxylum Fructus (Japanese pepper), Zingiberis Siccatum Rhizoma (processed ginger), Ginseng Radix (Asian ginseng), and maltose powder [10]. In the other three Kampo medicines besides DKT, several herbs are blended and, apparently, the role of sansho is relatively small. The reasons for combining different herbal drugs are not fully understood and are being scientifically investigated. Unlike other herbal drugs, DKT has been studied to some extent. 6-Shogaol and 6-gingerol in ginger have a warming effect on the stomach and intestine, and ginseng compounds, such as ginsenoside, are believed to be effective in relieving fatigue, weakness, anorexia, and indigestion. Sansho has been shown to promote the secretion of gastrointestinal hormones, such as motilin, which may be linked to the promotion of gastrointestinal motility [12]. These findings suggest that sansho present in DKT plays a major role in gastrointestinal motility.

  1. The statistical tools used in the analysis of the results is not mentioned in the materials and methodology section.

We have added a section on statistical analysis.

“2.7 Statistical analysis

All data are expressed as mean ± standard deviation (SD) for each group. Statistical analysis was performed using one-way analysis of variance with Student’s t-test. A value of P < 0.05 was considered to indicate statistical significance and is indicated using a single asterisk (*) or different letters.”

  1. Is there any possibility to elucidate the underline mechanism by anti-oxidants like gallic acid and quercetin to stabilize HαS? If yes, why it is not included?

We have added the following discussion in this regard in section 4 of the revised manuscript:

We believe that the mechanism by which antioxidants stabilize HαS is the same as that involved in the prevention of oxidation of vegetable oils rich in unsaturated fatty acids. In this case, the antioxidants do not bind to the unsaturated fatty acids at a specific site through a special binding, such as that seen for enzyme–substrate binding. The antioxidants are instead believed to react with oxygen radicals present in the environment surrounding the unsaturated fatty acid, reducing the frequency with which oxygen radicals approach the unsaturated fatty acid. Furthermore, as shown in Figure 8, high concentrations of gallic acid derivatives are required for stabilization of HαS. Quercetin is effective even at low concentrations; however, because it is not on the list of food additives as an antioxidant, we did not conduct further studies on the mechanism of action of gallic acid derivatives and quercetin. We intend to further investigate these mechanisms in a future study.

  1. At what quantity the antioxidants were mixed with HαS?

As mentioned in section 2.3, in this experiment, the concentration of HαS in the segment membrane solution was 2.5 mM, which was mixed with the same amount of the test substance solution; the final concentration was, therefore, 1.25 mM. The horizontal axes in Figures 6 and 7 show the final concentrations of the stabilizing substance.

  1. The possibility to have an alteration in the biological activities of HαS cannot be ruled out. In such case why the biological activities not estimated after stabilizing HαS with anti-oxidants?

We thank the reviewer for the pertinent question. As we mentioned in our response above, it appears that antioxidants do not bind to specific sites on HαS, but rather drift around HαS to scavenge the approaching oxygen radicals. Among antioxidants, α-tocopherol appears to be more effective because its amphiphilic nature allows it to approach HαS more easily. However, α-tocopherol does not appear to significantly alter the biological activity of HαS because it only approaches HαS, and not binds to it. For example, the addition of α-tocopherol did not change the taste of sansho. For this reason, we did not examine the changes in biological activity in this study.

  1. The introduction should also include small section on biological activities of hydroxy-α-sanshool.

We thank the reviewer for the valuable suggestion. We have added relevant information in the introduction section of the revised manuscript, as follows:

In addition to its gastrointestinal effects, strong salivating effects of HαS have been reported [13]. It is well known that HαS is involved for the activation of transient receptor potential cation channel subfamily V (TRPV) and activation of somatosensory neurons through the inhibition of two-pore potassium channels (KCNK3, KCNK9, and KCNK18) [14-16]. Local anesthetic effects of HαS have also been studied [9]. HαS has also been reported to improve lipid metabolism in animal studies using rats and mice [17,18]. HαS can potentially be used as a therapeutic and health-promoting agent. Therefore, for further pharmacological research and development of its medicinal uses, sanshools need to be stabilized.

August 09, 2023

Ms. Jessie Xu

Section Managing Editor

Foods

Dear Editor:

I wish to resubmit the manuscript titled “Stabilization of hydroxy-α-sanshool by antioxidants present in the genus Zanthoxylum.” The manuscript ID is Foods-2544044.

We double-checked that all references are relevant to the content of the manuscript.  Revisions to the manuscript are indicated in red. The cover letter includes a point-by-point explanation of the revisions made to the manuscript and the referee's response.

We thank you and the reviewers for your thoughtful suggestions and insights. The manuscript has benefited from these insightful suggestions. I look forward to working with you and the reviewers to move this manuscript closer to publication in Foods.

The manuscript has been rechecked and the necessary changes have been made in accordance with the reviewers’ suggestions. The responses to all comments have been prepared and attached herewith.

Thank you for your consideration. I look forward to hearing from you.

Sincerely,

Takahiko Mitani

Center of Regional Revitalization, Research Center for Food and Agriculture

Wakayama University, Wakayama

Wakayama 640-8510

Japan

Tel.: +81-73-457-7562

Reviewer 2 Report

Subject: Review of Manuscript (FOODS-2544044)

I am writing to provide my review of the manuscript titled “Stabilization of hydroxy-α-sanshool by antioxidants present in the genus Zanthoxylum” which has been submitted for publication in Foods (ISSN 2304-8158). I have carefully assessed the article and believe it requires substantial revisions.

The manuscript discusses the sensitivity of shansools and potential antioxidants to prevent oxidation. While the topic is relevant for foods, the overall quality of the manuscript needs improvement. 

The Introduction should be significantly enhanced to provide a clearer context and a comprehensive literature review to support the significance of the research.

Throughout the manuscript, there are various instances where numbers are referenced without corresponding explanations in the main body of the text. This inconsistency should be addressed to maintain clarity and cohesion in the article.

Furthermore, the article lacks essential details that are crucial for a scientific publication. For example, there is no information on the validation of the accelerated test used for determining oxidative stability. Its the major point for my decision. 

The manuscript also fails to mention the sample size used in the experiments, which is an essential aspect of the study design. Additionally, important analytical details are missing, such as the substances measured in HPLC and LC-MS analyses, LOD, LOQ, as well as the specific concentrations tested. These omissions hinder a comprehensive understanding of the research findings.

Given these concerns, it is my recommendation that the authors undertake significant revisions to address the mentioned issues and improve the overall quality of the article.  

Thank you for the opportunity to review this manuscript. I remain available for any further clarifications or assistance that may be required.

Author Response

The Introduction should be significantly enhanced to provide a clearer context and a comprehensive literature review to support the significance of the research.

We agree with your point and have rewritten it as shown in the file.

In particular, the biological functionality of HαS is appended in red in the Introduction.

Throughout the manuscript, there are various instances where numbers are referenced without corresponding explanations in the main body of the text. This inconsistency should be addressed to maintain clarity and cohesion in the article.

We apologize for the error in the citation. We have rechecked and corrected the error.

Furthermore, the article lacks essential details that are crucial for a scientific publication. For example, there is no information on the validation of the accelerated test used for determining oxidative stability. Its the major point for my decision.

Thank you for pointing it out. The assay method is very simple and reproducible. The important point is to keep the amount of HαS in the alcohol extract of segment membrane powder constant; the amount of HαS contained between lots of segment membrane varies in the range of ±2% of the average value. Therefore, in advance, we adjusted the amount of HαS in the extract solution by adding ethanol so that the amount of HαS in the extract solution would remain constant throughout the test. An equal amount of ethanol containing a stabilizer was added to this solution, and an accelerated test was conducted at 70°C. When ethanol without stabilizer was used, the amount of HαS remaining in the test solution was 61.35±1.18% in the mean ± SD value. If it was above this value, it was judged to have a stabilizing effect, and if it was below, it was judged to have no stabilizing effect or to destabilize HαS. Therefore, Section 2.3. was modified as follows.

2.3. Accelerated test of HαS-stabilizing activity

Three lots of sansho fruit were harvested from three sansho trees and dried separately. From dried pericarps of the three lots (2 kg each), 50 g of pericarps were used for segmental membrane separation. Segment membranes (12.5 g each) were obtained. Segment membranes from each lot were powdered as described above. To perform the assay for one candidate compound in triplicate, segment membranes powder (0.4 g) was suspended in ethanol (8 mL) and mixed by vortexing. The suspension was centrifuged at 2,000 x g for 2 min and the supernatant was collected. The amount of HαS in the alcohol extract of the segment membrane powder was determined by the HPLC method described below, and ethanol was added to bring the final HαS amount to 2.5 μmol/ml. Two ml of supernatant and 2 ml of the same volume of ethanol containing the candidate stabilizer were mixed in a 10 ml Spitz tube and incubated in an oil bath at 70°C. The amount of HαS in the sample on day 7 of incubation was determined using HPLC. The ratio of the amount of HαS on day 7 of incubation to the amount on day 0 was calculated and expressed as survival rate (%). When ethanol without stabilizer was used, the amount of HαS remaining in the test solution was 61.35±1.18% in the mean ± SD value. If it was above this value, it was judged to have a stabilizing effect, and if it was below, it was judged to have no stabilizing effect or to destabilize HαS.

The manuscript also fails to mention the sample size used in the experiments, which is an essential aspect of the study design. Additionally, important analytical details are missing, such as the substances measured in HPLC and LC-MS analyses, LOD, LOQ, as well as the specific concentrations tested. These omissions hinder a comprehensive understanding of the research findings.

In section 2.5., since HβS quantification is not used in this exam, we have removed the description. The quantitative method for HαS was rewritten. To illustrate a few points, HαS of the samples listed in Materials and methods were diluted to 0.38, 0.76, 1.90, and 3.8 mmole/ml, respectively, using ethanol solution containing 0.1% α-tocopherol, and analyzed by HPLC (n=3) under the conditions described above. In this concentration range, a linear relationship was obtained between the HαS concentration and the HPLC peak areas, with correlation coefficient of 0.9999. The amount of HαS in the sample was determined using each calibration curve. The experiment was also performed three times on different days, but no daily difference was found. The LOD was 0.00163 µmole/ml which was obtained from the standard deviation of the blank response value and the slope of the calibration curve.

In response to your suggestion, we have rewritten Section 2.5 as follows,

2.5. Quantitation of HαS

HαS of the samples listed in Materials and methods were diluted to 0.38, 0.76, 1.90, and 3.8 mmole/ml, respectively, using ethanol solution containing 0.1% α-Toc, and analyzed by HPLC (n=3) under the conditions described above. In this concentration range, a linear relationship was obtained between the HαS concentration and the HPLC peak areas. This calibration curve was used to determine the HαS content in the sample. The experiment was also performed three times on different days, but no daily difference was found. The LOD in this method was 0.00163 µmole/ml.

Given these concerns, it is my recommendation that the authors undertake significant revisions to address the mentioned issues and improve the overall quality of the article. 

We further reviewed and rewrote the entire paper as shown in manuscript ver.2.

Reviewer 3 Report

the Manuscript provides very valuable information however the manuscript is not well planned, designed and discussed.  It appears that the authors have put effort into this manuscript, however, it lacks the readability and clarity in the data presented.

I have the following comments for this manuscript:

  1. Please provide the details of HPLC and LC-MS experimental. In particular add more information about MS method.

2.     Please rewrite the sections:

Ø  Quantitation of HαS and HβS

Ø  Quantitation of tocopherol

      Because no useful information is reported. Add details.

3.     Provide the details of HPLC and LC-MS experimental. In particular add more information about MS method.

4.     Material: I don’t understand if only one sample was used. These data can be reproduced? Have you confirmed by sufficient replicates?

5.     In the section Results, in particular 3.2, add more information. The results are confusing, not detailed. According to me it would be needed to rewrite this section.  

Author Response

Please provide the details of HPLC and LC-MS experimental. In particular add more information about MS method.

  1. Please rewrite the sections:

Ø Quantitation of HαS

In section 2.5., since HβS quantification is not used in this exam, we have removed the description. The quantitative method for HαS was rewritten.

HαS of the samples listed in Materials and methods were diluted to 0.38, 0.76, 1.90, and 3.8 mmole/ml, respectively, using ethanol solution containing 0.1% α-tocopherol, and analyzed by HPLC (n=3) under the conditions described above. In this concentration range, a linear relationship was obtained between the HαS concentration and the HPLC peak areas, with correlation coefficient of 0.9999. The amount of HαS in the sample was determined using each calibration curve. The experiment was also performed three times on different days, but no daily difference was found. The LOD was 0.00163 µmole/ml which was obtained from the standard deviation of the blank response value and the slope of the calibration curve. 

In response to your suggestion, we have rewritten Section 2.5 as follows,

2.5. Quantitation of HαS

HαS of the samples listed in Materials and methods were diluted to 0.38, 0.76, 1.90, and 3.8 mmole/ml, respectively, using ethanol solution containing 0.1% α-Toc, and analyzed by HPLC (n=3) under the conditions described above. In this concentration range, a linear relationship was obtained between the HαS concentration and the HPLC peak areas. This calibration curve was used to determine the HαS content in the sample. The experiment was also performed three times on different days, but no daily difference was found. The LOD in this method was 0.00163 µmole/ml.

Ø Quantitation of tocopherol

Because no useful information is reported. Add details.

In response to your suggestion, we have rewritten it as follows:

2.6. Quantitation of tocopherol

Analysis of α-, β-, γ-, and δ-tocopherols was performed using HPLC by Japan Food Research Laboratories (Suita, Osaka, Japan). NaCl (1%, 1 mL), 6% pyrogallol in ethanol (10 mL), ethanol (1 mL) incorporating 2,2,5,7,8-pentamethyl-6-hydroxychroman (PMC; 10 mg), and 60% KOH (1.5 mL) were added to the crushed pericarps (0.5 g). The mixture was kept at 70 °C for 30 min. After cooling in an ice−water mixture, the unsaponified compound was extracted three times with 1% NaCl (15 mL) and 1:9 ethyl acetate/hexane (15 mL). The organic solvent layer was combined and evaporated to dryness under vacuum [20]. The residue was dissolved in hexane (1 mL) and analyzed by HPLC. HPLC was performed on a SHIMADZU LC-20AT instrument with fluorescence detector RF-10AXL (SHIMADZU, Kyoto, Japan). All samples were injected into an YMC-Pack SIL-06 S-5 μm ,4.6 ×150 mm column (YMC CO., LTD. Kyoto, Japan) for α and δ-Toc, and YMC-Pack SIL-06 S-5 μm, 4.6 mm×250 mm for β and γ-Toc. The analytes were eluted with acetic acid/n-hexane/2-propanol (5:100:2 v/v/v) at a flow rate of 1.5 mL/min at 40 °C. The excitation and emission wavelengths were 298 and 325 nm respectively.

  1. Provide the details of HPLC and LC-MS experimental. In particular add more information about MS method.

The following changes have been made.

2.4. HPLC and LC-MS

HPLC was performed on a SHIMADZU LC-2010 instrument (SHIMADZU, Kyoto, Japan). All samples were injected into an InertSustainSwift C18 4.6 × 250 mm column (GL Sciences, Tokyo, Japan). The analytes were eluted with mixed solvent A (30% acetonitrile): B (80% acetonitrile) at a flow rate of 1.0 mL/min. The conditions were as follows: 0 min: 0% B; 35 min: 45% B; 50 min: 100% B; 51 min: 0% B; 55 min: 0% B. All analyses were performed at 40 °C and the absorbance of eluate was monitored at 270 nm. Liquid chromatography–mass spectrometry (LC-MS) was performed by electrospray ionization (ESI)-MS in a positive mode using a Bruker micrOTOFTM ESI-TOF mass spectrometer (Bruker Corporation, Billerica, MA, USA) interfaced with an Agilent 1200 HPLC system (Agilent Technologies, Inc., Santa Clara, CA, USA). The MS conditions are follows: Capillary voltage, 4500V; Drying gas temperature, 200 ï‚°C; Drying gas flow, 8.0 L/min; Nebulizer pressure, 1.6 Bar. A portion of the sample was loaded onto an Inertsil ODS-3 (2.1 × 150 mm) column. Separation was achieved by elution with a mixture of solvents A (30% acetonitrile) and B (80% acetonitrile) at a flow rate of 1.0 mL/min. The conditions were as follows: 0 min: 0% B; 35 min: 45% B; 50 min: 100% B; 51 min: 0% B; 55 min: 0% B. All analyses were performed at 40 °C and the absorbance of eluate was monitored at 270 nm.

  1. Material: I don’t understand if only one sample was used. These data can be reproduced? Have you confirmed by sufficient replicates?

 Thank you for pointing it out. We have rewritten Section 2.3. as follows:

2.3. Accelerated test of HαS-stabilizing activity

Three lots of sansho fruit were harvested from three sansho trees and dried separately. From dried pericarps of the three lots (2 kg each), 50 g of pericarps were used for segmental membrane separation. Segmental membranes (12.5 g each) were obtained. Segmental membranes from each lot were powdered as described above. To perform the assay for one candidate compound in triplicate, segmental membranes powder (0.4 g) was suspended in ethanol (8 mL) and mixed by vortexing. The suspension was centrifuged at 2,000 x g for 2 min and the supernatant was collected. The amount of HαS in the alcohol extract of the segmental membrane powder was determined by the HPLC method described below, and ethanol was added to bring the final HαS amount to 2.5 μmol/ml. Two ml of supernatant and 2 ml of the same volume of ethanol containing the candidate stabilizer were mixed in a 10 ml Spitz tube and incubated in an oil bath at 70°C. The amount of HαS in the sample on day 7 of incubation was determined using HPLC. The ratio of the amount of HαS on day 7 of incubation to the amount on day 0 was calculated and expressed as survival rate (%). When ethanol without stabilizer was used, the amount of HαS remaining in the test solution was 61.35±1.18% in the mean ± SD value. If it was above this value, it was judged to have a stabilizing effect, and if it was below, it was judged to have no stabilizing effect or to destabilize HαS.

  1. In the section Results, in particular 3.2, add more information. The results are confusing, not detailed. According to me it would be needed to rewrite this section.

Thank you for pointing it out. We have rewritten Section 3.2. as follows:

3.2. HPLC and LC/MS of ethanol extracts of pericarps

A chromatogram of the ethanolic extract of pericarp is shown in Figure 2a. Several peaks were observed in the chromatogram. Of these, peaks A, B, C, and D were found to be of HεS, HαS, HβS, and HγS respectively, by comparison with the standard sanshools. In addition, MS analysis was performed for peaks B, E, and F. C16H25NO2 due to the signals of 264.1958 (M+H+) and 286.1778 (M+Na+) based on high-resolution LC-MS (ESI, positive mode) data. The LC-MS data for peaks E and F indicated elemental compositions of C16H26NO (M+H+) and C18H28NO (M+H+), respectively. The molecular weights of the three major peaks B, E, and F were 263.1958, 247.3758, and 273.4131, respectively. Peaks B, E, and F were believed to be for HαS, α-sanshool (αS), and γ-sanshool (γS). From HPLC of references performed under the similar conditions [14,21,22], we assumed that peaks E and F might be for αS and γS, respectively.

Please provide the details of HPLC and LC-MS experimental. In particular add more information about MS method.

  1. Please rewrite the sections:

Ø Quantitation of HαS

In section 2.5., since HβS quantification is not used in this exam, we have removed the description. The quantitative method for HαS was rewritten.

HαS of the samples listed in Materials and methods were diluted to 0.38, 0.76, 1.90, and 3.8 mmole/ml, respectively, using ethanol solution containing 0.1% α-tocopherol, and analyzed by HPLC (n=3) under the conditions described above. In this concentration range, a linear relationship was obtained between the HαS concentration and the HPLC peak areas, with correlation coefficient of 0.9999. The amount of HαS in the sample was determined using each calibration curve. The experiment was also performed three times on different days, but no daily difference was found. The LOD was 0.00163 µmole/ml which was obtained from the standard deviation of the blank response value and the slope of the calibration curve. 

In response to your suggestion, we have rewritten Section 2.5 as follows,

2.5. Quantitation of HαS

HαS of the samples listed in Materials and methods were diluted to 0.38, 0.76, 1.90, and 3.8 mmole/ml, respectively, using ethanol solution containing 0.1% α-Toc, and analyzed by HPLC (n=3) under the conditions described above. In this concentration range, a linear relationship was obtained between the HαS concentration and the HPLC peak areas. This calibration curve was used to determine the HαS content in the sample. The experiment was also performed three times on different days, but no daily difference was found. The LOD in this method was 0.00163 µmole/ml.

Ø Quantitation of tocopherol

Because no useful information is reported. Add details.

In response to your suggestion, we have rewritten it as follows:

2.6. Quantitation of tocopherol

Analysis of α-, β-, γ-, and δ-tocopherols was performed using HPLC by Japan Food Research Laboratories (Suita, Osaka, Japan). NaCl (1%, 1 mL), 6% pyrogallol in ethanol (10 mL), ethanol (1 mL) incorporating 2,2,5,7,8-pentamethyl-6-hydroxychroman (PMC; 10 mg), and 60% KOH (1.5 mL) were added to the crushed pericarps (0.5 g). The mixture was kept at 70 °C for 30 min. After cooling in an ice−water mixture, the unsaponified compound was extracted three times with 1% NaCl (15 mL) and 1:9 ethyl acetate/hexane (15 mL). The organic solvent layer was combined and evaporated to dryness under vacuum [20]. The residue was dissolved in hexane (1 mL) and analyzed by HPLC. HPLC was performed on a SHIMADZU LC-20AT instrument with fluorescence detector RF-10AXL (SHIMADZU, Kyoto, Japan). All samples were injected into an YMC-Pack SIL-06 S-5 μm ,4.6 ×150 mm column (YMC CO., LTD. Kyoto, Japan) for α and δ-Toc, and YMC-Pack SIL-06 S-5 μm, 4.6 mm×250 mm for β and γ-Toc. The analytes were eluted with acetic acid/n-hexane/2-propanol (5:100:2 v/v/v) at a flow rate of 1.5 mL/min at 40 °C. The excitation and emission wavelengths were 298 and 325 nm respectively.

  1. Provide the details of HPLC and LC-MS experimental. In particular add more information about MS method.

The following changes have been made.

2.4. HPLC and LC-MS

HPLC was performed on a SHIMADZU LC-2010 instrument (SHIMADZU, Kyoto, Japan). All samples were injected into an InertSustainSwift C18 4.6 × 250 mm column (GL Sciences, Tokyo, Japan). The analytes were eluted with mixed solvent A (30% acetonitrile): B (80% acetonitrile) at a flow rate of 1.0 mL/min. The conditions were as follows: 0 min: 0% B; 35 min: 45% B; 50 min: 100% B; 51 min: 0% B; 55 min: 0% B. All analyses were performed at 40 °C and the absorbance of eluate was monitored at 270 nm. Liquid chromatography–mass spectrometry (LC-MS) was performed by electrospray ionization (ESI)-MS in a positive mode using a Bruker micrOTOFTM ESI-TOF mass spectrometer (Bruker Corporation, Billerica, MA, USA) interfaced with an Agilent 1200 HPLC system (Agilent Technologies, Inc., Santa Clara, CA, USA). The MS conditions are follows: Capillary voltage, 4500V; Drying gas temperature, 200 ï‚°C; Drying gas flow, 8.0 L/min; Nebulizer pressure, 1.6 Bar. A portion of the sample was loaded onto an Inertsil ODS-3 (2.1 × 150 mm) column. Separation was achieved by elution with a mixture of solvents A (30% acetonitrile) and B (80% acetonitrile) at a flow rate of 1.0 mL/min. The conditions were as follows: 0 min: 0% B; 35 min: 45% B; 50 min: 100% B; 51 min: 0% B; 55 min: 0% B. All analyses were performed at 40 °C and the absorbance of eluate was monitored at 270 nm.

  1. Material: I don’t understand if only one sample was used. These data can be reproduced? Have you confirmed by sufficient replicates?

 Thank you for pointing it out. We have rewritten Section 2.3. as follows:

2.3. Accelerated test of HαS-stabilizing activity

Three lots of sansho fruit were harvested from three sansho trees and dried separately. From dried pericarps of the three lots (2 kg each), 50 g of pericarps were used for segmental membrane separation. Segmental membranes (12.5 g each) were obtained. Segmental membranes from each lot were powdered as described above. To perform the assay for one candidate compound in triplicate, segmental membranes powder (0.4 g) was suspended in ethanol (8 mL) and mixed by vortexing. The suspension was centrifuged at 2,000 x g for 2 min and the supernatant was collected. The amount of HαS in the alcohol extract of the segmental membrane powder was determined by the HPLC method described below, and ethanol was added to bring the final HαS amount to 2.5 μmol/ml. Two ml of supernatant and 2 ml of the same volume of ethanol containing the candidate stabilizer were mixed in a 10 ml Spitz tube and incubated in an oil bath at 70°C. The amount of HαS in the sample on day 7 of incubation was determined using HPLC. The ratio of the amount of HαS on day 7 of incubation to the amount on day 0 was calculated and expressed as survival rate (%). When ethanol without stabilizer was used, the amount of HαS remaining in the test solution was 61.35±1.18% in the mean ± SD value. If it was above this value, it was judged to have a stabilizing effect, and if it was below, it was judged to have no stabilizing effect or to destabilize HαS.

  1. In the section Results, in particular 3.2, add more information. The results are confusing, not detailed. According to me it would be needed to rewrite this section.

Thank you for pointing it out. We have rewritten Section 3.2. as follows:

3.2. HPLC and LC/MS of ethanol extracts of pericarps

A chromatogram of the ethanolic extract of pericarp is shown in Figure 2a. Several peaks were observed in the chromatogram. Of these, peaks A, B, C, and D were found to be of HεS, HαS, HβS, and HγS respectively, by comparison with the standard sanshools. In addition, MS analysis was performed for peaks B, E, and F. C16H25NO2 due to the signals of 264.1958 (M+H+) and 286.1778 (M+Na+) based on high-resolution LC-MS (ESI, positive mode) data. The LC-MS data for peaks E and F indicated elemental compositions of C16H26NO (M+H+) and C18H28NO (M+H+), respectively. The molecular weights of the three major peaks B, E, and F were 263.1958, 247.3758, and 273.4131, respectively. Peaks B, E, and F were believed to be for HαS, α-sanshool (αS), and γ-sanshool (γS). From HPLC of references performed under the similar conditions [14,21,22], we assumed that peaks E and F might be for αS and γS, respectively.

Please provide the details of HPLC and LC-MS experimental. In particular add more information about MS method.

  1. Please rewrite the sections:

Ø Quantitation of HαS

In section 2.5., since HβS quantification is not used in this exam, we have removed the description. The quantitative method for HαS was rewritten.

HαS of the samples listed in Materials and methods were diluted to 0.38, 0.76, 1.90, and 3.8 mmole/ml, respectively, using ethanol solution containing 0.1% α-tocopherol, and analyzed by HPLC (n=3) under the conditions described above. In this concentration range, a linear relationship was obtained between the HαS concentration and the HPLC peak areas, with correlation coefficient of 0.9999. The amount of HαS in the sample was determined using each calibration curve. The experiment was also performed three times on different days, but no daily difference was found. The LOD was 0.00163 µmole/ml which was obtained from the standard deviation of the blank response value and the slope of the calibration curve. 

In response to your suggestion, we have rewritten Section 2.5 as follows,

2.5. Quantitation of HαS

HαS of the samples listed in Materials and methods were diluted to 0.38, 0.76, 1.90, and 3.8 mmole/ml, respectively, using ethanol solution containing 0.1% α-Toc, and analyzed by HPLC (n=3) under the conditions described above. In this concentration range, a linear relationship was obtained between the HαS concentration and the HPLC peak areas. This calibration curve was used to determine the HαS content in the sample. The experiment was also performed three times on different days, but no daily difference was found. The LOD in this method was 0.00163 µmole/ml.

Ø Quantitation of tocopherol

Because no useful information is reported. Add details.

In response to your suggestion, we have rewritten it as follows:

2.6. Quantitation of tocopherol

Analysis of α-, β-, γ-, and δ-tocopherols was performed using HPLC by Japan Food Research Laboratories (Suita, Osaka, Japan). NaCl (1%, 1 mL), 6% pyrogallol in ethanol (10 mL), ethanol (1 mL) incorporating 2,2,5,7,8-pentamethyl-6-hydroxychroman (PMC; 10 mg), and 60% KOH (1.5 mL) were added to the crushed pericarps (0.5 g). The mixture was kept at 70 °C for 30 min. After cooling in an ice−water mixture, the unsaponified compound was extracted three times with 1% NaCl (15 mL) and 1:9 ethyl acetate/hexane (15 mL). The organic solvent layer was combined and evaporated to dryness under vacuum [20]. The residue was dissolved in hexane (1 mL) and analyzed by HPLC. HPLC was performed on a SHIMADZU LC-20AT instrument with fluorescence detector RF-10AXL (SHIMADZU, Kyoto, Japan). All samples were injected into an YMC-Pack SIL-06 S-5 μm ,4.6 ×150 mm column (YMC CO., LTD. Kyoto, Japan) for α and δ-Toc, and YMC-Pack SIL-06 S-5 μm, 4.6 mm×250 mm for β and γ-Toc. The analytes were eluted with acetic acid/n-hexane/2-propanol (5:100:2 v/v/v) at a flow rate of 1.5 mL/min at 40 °C. The excitation and emission wavelengths were 298 and 325 nm respectively.

  1. Provide the details of HPLC and LC-MS experimental. In particular add more information about MS method.

The following changes have been made.

2.4. HPLC and LC-MS

HPLC was performed on a SHIMADZU LC-2010 instrument (SHIMADZU, Kyoto, Japan). All samples were injected into an InertSustainSwift C18 4.6 × 250 mm column (GL Sciences, Tokyo, Japan). The analytes were eluted with mixed solvent A (30% acetonitrile): B (80% acetonitrile) at a flow rate of 1.0 mL/min. The conditions were as follows: 0 min: 0% B; 35 min: 45% B; 50 min: 100% B; 51 min: 0% B; 55 min: 0% B. All analyses were performed at 40 °C and the absorbance of eluate was monitored at 270 nm. Liquid chromatography–mass spectrometry (LC-MS) was performed by electrospray ionization (ESI)-MS in a positive mode using a Bruker micrOTOFTM ESI-TOF mass spectrometer (Bruker Corporation, Billerica, MA, USA) interfaced with an Agilent 1200 HPLC system (Agilent Technologies, Inc., Santa Clara, CA, USA). The MS conditions are follows: Capillary voltage, 4500V; Drying gas temperature, 200 ï‚°C; Drying gas flow, 8.0 L/min; Nebulizer pressure, 1.6 Bar. A portion of the sample was loaded onto an Inertsil ODS-3 (2.1 × 150 mm) column. Separation was achieved by elution with a mixture of solvents A (30% acetonitrile) and B (80% acetonitrile) at a flow rate of 1.0 mL/min. The conditions were as follows: 0 min: 0% B; 35 min: 45% B; 50 min: 100% B; 51 min: 0% B; 55 min: 0% B. All analyses were performed at 40 °C and the absorbance of eluate was monitored at 270 nm.

  1. Material: I don’t understand if only one sample was used. These data can be reproduced? Have you confirmed by sufficient replicates?

 Thank you for pointing it out. We have rewritten Section 2.3. as follows:

2.3. Accelerated test of HαS-stabilizing activity

Three lots of sansho fruit were harvested from three sansho trees and dried separately. From dried pericarps of the three lots (2 kg each), 50 g of pericarps were used for segmental membrane separation. Segmental membranes (12.5 g each) were obtained. Segmental membranes from each lot were powdered as described above. To perform the assay for one candidate compound in triplicate, segmental membranes powder (0.4 g) was suspended in ethanol (8 mL) and mixed by vortexing. The suspension was centrifuged at 2,000 x g for 2 min and the supernatant was collected. The amount of HαS in the alcohol extract of the segmental membrane powder was determined by the HPLC method described below, and ethanol was added to bring the final HαS amount to 2.5 μmol/ml. Two ml of supernatant and 2 ml of the same volume of ethanol containing the candidate stabilizer were mixed in a 10 ml Spitz tube and incubated in an oil bath at 70°C. The amount of HαS in the sample on day 7 of incubation was determined using HPLC. The ratio of the amount of HαS on day 7 of incubation to the amount on day 0 was calculated and expressed as survival rate (%). When ethanol without stabilizer was used, the amount of HαS remaining in the test solution was 61.35±1.18% in the mean ± SD value. If it was above this value, it was judged to have a stabilizing effect, and if it was below, it was judged to have no stabilizing effect or to destabilize HαS.

  1. In the section Results, in particular 3.2, add more information. The results are confusing, not detailed. According to me it would be needed to rewrite this section.

Thank you for pointing it out. We have rewritten Section 3.2. as follows:

3.2. HPLC and LC/MS of ethanol extracts of pericarps

A chromatogram of the ethanolic extract of pericarp is shown in Figure 2a. Several peaks were observed in the chromatogram. Of these, peaks A, B, C, and D were found to be of HεS, HαS, HβS, and HγS respectively, by comparison with the standard sanshools. In addition, MS analysis was performed for peaks B, E, and F. C16H25NO2 due to the signals of 264.1958 (M+H+) and 286.1778 (M+Na+) based on high-resolution LC-MS (ESI, positive mode) data. The LC-MS data for peaks E and F indicated elemental compositions of C16H26NO (M+H+) and C18H28NO (M+H+), respectively. The molecular weights of the three major peaks B, E, and F were 263.1958, 247.3758, and 273.4131, respectively. Peaks B, E, and F were believed to be for HαS, α-sanshool (αS), and γ-sanshool (γS). From HPLC of references performed under the similar conditions [14,21,22], we assumed that peaks E and F might be for αS and γS, respectively.

Please provide the details of HPLC and LC-MS experimental. In particular add more information about MS method.

  1. Please rewrite the sections:

Ø Quantitation of HαS

In section 2.5., since HβS quantification is not used in this exam, we have removed the description. The quantitative method for HαS was rewritten.

HαS of the samples listed in Materials and methods were diluted to 0.38, 0.76, 1.90, and 3.8 mmole/ml, respectively, using ethanol solution containing 0.1% α-tocopherol, and analyzed by HPLC (n=3) under the conditions described above. In this concentration range, a linear relationship was obtained between the HαS concentration and the HPLC peak areas, with correlation coefficient of 0.9999. The amount of HαS in the sample was determined using each calibration curve. The experiment was also performed three times on different days, but no daily difference was found. The LOD was 0.00163 µmole/ml which was obtained from the standard deviation of the blank response value and the slope of the calibration curve. 

In response to your suggestion, we have rewritten Section 2.5 as follows,

2.5. Quantitation of HαS

HαS of the samples listed in Materials and methods were diluted to 0.38, 0.76, 1.90, and 3.8 mmole/ml, respectively, using ethanol solution containing 0.1% α-Toc, and analyzed by HPLC (n=3) under the conditions described above. In this concentration range, a linear relationship was obtained between the HαS concentration and the HPLC peak areas. This calibration curve was used to determine the HαS content in the sample. The experiment was also performed three times on different days, but no daily difference was found. The LOD in this method was 0.00163 µmole/ml.

Ø Quantitation of tocopherol

Because no useful information is reported. Add details.

In response to your suggestion, we have rewritten it as follows:

2.6. Quantitation of tocopherol

Analysis of α-, β-, γ-, and δ-tocopherols was performed using HPLC by Japan Food Research Laboratories (Suita, Osaka, Japan). NaCl (1%, 1 mL), 6% pyrogallol in ethanol (10 mL), ethanol (1 mL) incorporating 2,2,5,7,8-pentamethyl-6-hydroxychroman (PMC; 10 mg), and 60% KOH (1.5 mL) were added to the crushed pericarps (0.5 g). The mixture was kept at 70 °C for 30 min. After cooling in an ice−water mixture, the unsaponified compound was extracted three times with 1% NaCl (15 mL) and 1:9 ethyl acetate/hexane (15 mL). The organic solvent layer was combined and evaporated to dryness under vacuum [20]. The residue was dissolved in hexane (1 mL) and analyzed by HPLC. HPLC was performed on a SHIMADZU LC-20AT instrument with fluorescence detector RF-10AXL (SHIMADZU, Kyoto, Japan). All samples were injected into an YMC-Pack SIL-06 S-5 μm ,4.6 ×150 mm column (YMC CO., LTD. Kyoto, Japan) for α and δ-Toc, and YMC-Pack SIL-06 S-5 μm, 4.6 mm×250 mm for β and γ-Toc. The analytes were eluted with acetic acid/n-hexane/2-propanol (5:100:2 v/v/v) at a flow rate of 1.5 mL/min at 40 °C. The excitation and emission wavelengths were 298 and 325 nm respectively.

  1. Provide the details of HPLC and LC-MS experimental. In particular add more information about MS method.

The following changes have been made.

2.4. HPLC and LC-MS

HPLC was performed on a SHIMADZU LC-2010 instrument (SHIMADZU, Kyoto, Japan). All samples were injected into an InertSustainSwift C18 4.6 × 250 mm column (GL Sciences, Tokyo, Japan). The analytes were eluted with mixed solvent A (30% acetonitrile): B (80% acetonitrile) at a flow rate of 1.0 mL/min. The conditions were as follows: 0 min: 0% B; 35 min: 45% B; 50 min: 100% B; 51 min: 0% B; 55 min: 0% B. All analyses were performed at 40 °C and the absorbance of eluate was monitored at 270 nm. Liquid chromatography–mass spectrometry (LC-MS) was performed by electrospray ionization (ESI)-MS in a positive mode using a Bruker micrOTOFTM ESI-TOF mass spectrometer (Bruker Corporation, Billerica, MA, USA) interfaced with an Agilent 1200 HPLC system (Agilent Technologies, Inc., Santa Clara, CA, USA). The MS conditions are follows: Capillary voltage, 4500V; Drying gas temperature, 200 ï‚°C; Drying gas flow, 8.0 L/min; Nebulizer pressure, 1.6 Bar. A portion of the sample was loaded onto an Inertsil ODS-3 (2.1 × 150 mm) column. Separation was achieved by elution with a mixture of solvents A (30% acetonitrile) and B (80% acetonitrile) at a flow rate of 1.0 mL/min. The conditions were as follows: 0 min: 0% B; 35 min: 45% B; 50 min: 100% B; 51 min: 0% B; 55 min: 0% B. All analyses were performed at 40 °C and the absorbance of eluate was monitored at 270 nm.

  1. Material: I don’t understand if only one sample was used. These data can be reproduced? Have you confirmed by sufficient replicates?

 Thank you for pointing it out. We have rewritten Section 2.3. as follows:

2.3. Accelerated test of HαS-stabilizing activity

Three lots of sansho fruit were harvested from three sansho trees and dried separately. From dried pericarps of the three lots (2 kg each), 50 g of pericarps were used for segmental membrane separation. Segmental membranes (12.5 g each) were obtained. Segmental membranes from each lot were powdered as described above. To perform the assay for one candidate compound in triplicate, segmental membranes powder (0.4 g) was suspended in ethanol (8 mL) and mixed by vortexing. The suspension was centrifuged at 2,000 x g for 2 min and the supernatant was collected. The amount of HαS in the alcohol extract of the segmental membrane powder was determined by the HPLC method described below, and ethanol was added to bring the final HαS amount to 2.5 μmol/ml. Two ml of supernatant and 2 ml of the same volume of ethanol containing the candidate stabilizer were mixed in a 10 ml Spitz tube and incubated in an oil bath at 70°C. The amount of HαS in the sample on day 7 of incubation was determined using HPLC. The ratio of the amount of HαS on day 7 of incubation to the amount on day 0 was calculated and expressed as survival rate (%). When ethanol without stabilizer was used, the amount of HαS remaining in the test solution was 61.35±1.18% in the mean ± SD value. If it was above this value, it was judged to have a stabilizing effect, and if it was below, it was judged to have no stabilizing effect or to destabilize HαS.

  1. In the section Results, in particular 3.2, add more information. The results are confusing, not detailed. According to me it would be needed to rewrite this section.

Thank you for pointing it out. We have rewritten Section 3.2. as follows:

3.2. HPLC and LC/MS of ethanol extracts of pericarps

A chromatogram of the ethanolic extract of pericarp is shown in Figure 2a. Several peaks were observed in the chromatogram. Of these, peaks A, B, C, and D were found to be of HεS, HαS, HβS, and HγS respectively, by comparison with the standard sanshools. In addition, MS analysis was performed for peaks B, E, and F. C16H25NO2 due to the signals of 264.1958 (M+H+) and 286.1778 (M+Na+) based on high-resolution LC-MS (ESI, positive mode) data. The LC-MS data for peaks E and F indicated elemental compositions of C16H26NO (M+H+) and C18H28NO (M+H+), respectively. The molecular weights of the three major peaks B, E, and F were 263.1958, 247.3758, and 273.4131, respectively. Peaks B, E, and F were believed to be for HαS, α-sanshool (αS), and γ-sanshool (γS). From HPLC of references performed under the similar conditions [14,21,22], we assumed that peaks E and F might be for αS and γS, respectively.

Reviewer 4 Report

Please see attached a few recommendations of the reviewer 

Author Response

Introduction

- Line-39: I suggest adding a couple of references from the literature here to back up the statements on the Japanese peppers.

Production of sansho in Japanis is extremely low, at about 700 tons wet weight in the most productive year. Therefore, there is no appropriate paper written in English. The following article, written in Japanese (No abstract written in English.), has been newly added. On the other hand, the production of Chinese Prickly Ash is 360,000 tons, which cannot be compared to the production scale of sansho.

  1. Dai, I. Fujii, K. Tsuji. Sansho production areas in Wakayama prefecture (in Japanese). In History of Agricultural Development in Wakayama Prefecture â…¡, Research Center for Food and Agriculture, Wakayama University, 2020; pp. 257-282.

Material Methods

- Sections 2.2-2.6:

Have you developed the methodology at your laboratory? If yes, please explicitly mention this…otherwise please give literature references/sources for each used method..

We have developed it at our laboratory. So, we wrote it at the end of Introduction.

- Please add at the end of the section a paragraph (2.7) on the statistical analysis of the results…did you use any ANOVA test? How did you come with statistically significant differences (e.g. figure 8?)

We have added 2.7 Statistical Analysis to Materials and Methods.

2.7 Statistical analysis

All data are expressed as the mean ± standard deviation (SD) values for each group. Statistical analysis was performed using one-way analysis of variance with Student’s t-test. P < 0.05 and are indicated by a single asterisk (*) or different letters.

Results

- Figure-7 is too much “concentrated” …not easy to distinguish the antioxidant effects.. I would suggest to replace with a table, instead, and indicate the statistical differences.

Thank you for pointing this out. At first, we also tried to make a table, but the table became complicated and troublesome. (We had first added the candidate stabilizers in % concentrations. When we later recalculated them to mM, the concentrations of each stabilizer were written in different ways.) We thought it was important to show at a glance that phenolic compounds require high concentrations to stabilize HαS (Figure 7), while α-tocopherol stabilizes HαS even at low concentrations (Figure 6). When performed by the t-test, statistical differences are likely to be observed, and we thought that it was more important to recognize the concentration range of stabilizing substances than to emphasize this.

Hence, we would like Figure 7 (changed to Figure 8) to remain as it is.

- 237-239: If beta-carotene not soluble in ethanol why selected as carotenoid example here? Why you did not select other carotenoids with polar groups (e.g. lutein, capsanthin etc) that may have exhibited a better solubility?

Thank you for pointing this out. Lutein is slightly soluble in ethanol and capsanthin is soluble in ethanol. We did not test these carotenoids because they are extremely expensive. Even if they were effective, we believe it is highly unlikely that they would be put into practical use as stabilizers for sanshools for price reasons. Therefore, we have added the following text to the manuscript.

Carotenoids are expected to be present in the pericarps of sansho and may contribute to the stabilization of sanshools. However, it is difficult to dissolve in ethanol, and even if it did dissolve, it is unstable by itself, making its application in food products difficult. This is an issue for the future.

Discussion-Conclusions

- Lines 326-359….I don’t understand how the given info link to the results…For me most of presented info here would be more suitable for the introduction section…

We agree with your point and have rewritten it as shown in the file.

Some text has been moved to the Introduction (in blue).

- I would suggest some restructuring-reformulation.. maybe results & discussion sections could merge…and then have a better structured conclusion section where to present in a couple of bullet points what are the most important findings of the manuscript/any follow up work and market applications?

In response to your suggestion, we have made the following statement.

We constructed a system for stabilizing substances in sanshools and found that α-Toc exhibits excellent activity. Stabilization of sanshools can be employed for developing new sansho products that exploit their excellent functionality.

However, since sanshools and α-Toc are not strongly bonded, they are likely to separate easily when made into liquids or powders. In the future, it will be necessary to develop a technology to make sanshools and α-Toc always coexist in close proximity.

Round 2

Reviewer 3 Report

The authors have responded by making the requested corrections. However, section 3.2, which was supposed to be integrated, has not been modified. Therefore, kindly add the requested content. Thank you.
